

# Endocranial anatomy of the ceratopsid dinosaur *Triceratops* and interpretations of sensory and motor function

Rina Sakagami[1] and Soichiro Kawabe[2,3]

[1] Department of Bioscience and Biotechnology, Fukui Prefectural University, Yoshida-gun, Eiheiji-cho, Fukui, Japan
[2] Institute of Dinosaur Research, Fukui Prefectural University, Yoshida-gun, Eiheiji-cho, Fukui, Japan
[3] Fukui Prefectural Dinosaur Museum, Katsuyama, Fukui, Japan

## ABSTRACT

*Triceratops* is one of the well-known Cretaceous ceratopsian dinosaurs. The ecology of *Triceratops* has been controversial because of its unique morphological features. However, arguments based on brain and inner ear structures have been scarce. In this study, two braincases (FPDM-V-9677 and FPDM-V-9775) were analyzed with computed tomography to generate three-dimensional virtual renderings of the endocasts of the cranial cavities and bony labyrinths. Quantitative analysis, including comparison of linear measurements of the degree of development of the olfactory bulb and inner ear, was performed on these virtual endocasts to acquire detailed neuroanatomical information. When compared with other dinosaurs, the olfactory bulb of *Triceratops* is relatively small, indicating that *Triceratops* had a reduced acuity in sense of smell. The lateral semicircular canal reveals that the basicranial axis of *Triceratops* is approximately 45° to the ground, which is an effective angle to display their horns as well as frill, and to graze. The semicircular canals of *Triceratops* are relatively smaller than those of primitive ceratopsians, such as *Psittacosaurus* and *Protoceratops*, suggesting that sensory input for the reflexive stabilization of gaze and posture of *Triceratops* was less developed than that of primitive ceratopsians. The cochlear length of *Triceratops* is relatively short when compared with other dinosaurs. Because cochlear length correlates with hearing frequency, *Triceratops* was likely adapted to hearing low frequencies.

## INTRODUCTION

A number of research works have discussed the sensorineural function of some ceratopsians based on neuroanatomical characteristics (*Brown, 1914*; *Brown & Schlaikjer, 1940*; *Hopson, 1979*; *Forster, 1996*; *Zhou et al., 2007*; *Witmer & Ridgely, 2008*). In particular, computed tomography (CT) has been used to capture such characteristics more accurately. For example, previous studies analyzed the skulls of *Psittacosaurus* using CT and reconstructed their virtual cranial endocast (*Zhou et al., 2007*; *Bullar et al., 2019*; *Napoli et al., 2019*). They observed various sensorineural anatomical features of *Psittacosaurus* on the basis of virtual endocasts. *Bullar et al. (2019)* compared the inner ear morphology of three specimens of *P. lujiatunensis* and discussed that their head posture changes with growth.

Corresponding author
Rina Sakagami,
sakagami.rina@gmail.com

*Napoli et al. (2019)* described the brain endocast and endosseous labyrinth of *P. amitabha*. *Zhou et al. (2007)* found that *Psittacosaurus* had tall vertical semicircular canals, which are larger than those of *Protoceratops* and Ceratopsidae. Because large curvature of the canals is consistent with, and indicative of, the ability to stabilize the gaze or locomotion maneuverability in some mammals and archosaurs, including birds (*Turkewitsch, 1934*; *Spoor & Zonneveld, 1998*; *Spoor et al., 2002*; *Witmer et al., 2008*; *Spoor et al., 2007*), it was inferred that *Psittacosaurus* was more agile than neoceratopsians. Additionally, *Zhou et al. (2007)* described *Psittacosaurus* as having a relatively short cochlear duct and suggested that hearing in *Psittacosaurus* was limited to lower frequencies. They estimated further that *Psittacosaurus* had large olfactory bulbs and argued that they had an acute sense of smell.

The endocranial structures of a basal neoceratopsian have also been elucidated in recent years, in which *Zhang et al. (2019)* illustrated the endocranial structures of *Auroraceratops*. The olfactory bulbs of *Auroraceratops* are larger than that of more derived ceratopsian *Pachyrhinosaurus* but is similar to that of *Pachyrhinosaurus* in that it is laterally wide and flat. They concluded that *Auroraceratops* had a keener sense of smell than *Pachyrhinosaurus*. They also found that the semicircular canals of *Auroraceratops* are slender and curved, being similar to that of *Psittacosaurus*. However, it differs from those of more derived neoceratopsians such as *Pachyrhinosaurus* and *Anchiceratops* with thick and short canals. Although they have made such morphological comparisons, they have not discussed the ecology of basal neoceratopsians based on inner ear morphology.

For the centrosaurine *Pachyrhinosaurus*, the brain endocasts of *P. canadensis* (*Langston Jr, 1975*), *P. lakustai* (*Witmer & Ridgely, 2008*) and *P. perotorum* (*Tykoski & Fiorillo, 2013*) have been described. *Witmer & Ridgely (2008)* described a virtual endocast of *P. lakustai* and discussed their paleoecology. Specifically, they found that *Pachyrhinosaurus* possesses more elongate semicircular canals than some other neoceratopsians. Elongation of semicircular canals is linked to coordinating eye movements and head rotation (*Spoor et al., 2007*). The authors considered that this elongation may have aided this species to gaze more steadily in comparison to some other neoceratopsians (*Witmer & Ridgely, 2008*). In addition, *Pachyrhinosaurus* has a short cochlear duct, suggesting that exceptional hearing sense was not important for *Pachyrhinosaurus*. *Witmer & Ridgely (2008)* also described the small olfactory bulbs, cerebrum, optic tecta, and cerebellum in *Pachyrhinosaurus*, concluding that the smallness of these regions of the brain may suggest that precise sensory integration and control were of lesser importance for *Pachyrhinosaurus*.

The natural endocast of *Anchiceratops* has been reported as an example within chasmosaurines (*Brown, 1914*). Later, *Zhou et al. (2007)* mentioned that the anterior semicircular canals (ASC) seen in *Anchiceratops* could be considered broadly similar to the posterior semicircular canal (PSC) in curvature and dimensions. A previous study reported that in humans and birds, more agile or high-degree maneuverable species have larger semicircular canals than that of slow-moving species (*Spoor & Zonneveld, 1998*). From this view, *Anchiceratops* was considered less agile than *Psittacosaurus*.

These previous studies assessed the sensorineural attributes and endocranial capabilities of ceratopsians, but none conducted quantitative evaluations. Moreover, although natural endocasts, latex or plaster casts of the cranial cavity of *Triceratops* have also been described

by some authors (*Bürckhardt, 1892*; *Marsh, 1896*; *Hay, 1909*; *Gilmore, 1919*; *Forster, 1996*; *Erickson, 2017*), detailed examination of the sensorineural function of *Triceratops* based on virtual endocasts has never been undertaken. Quantitative comparison of endocasts, including those of *Triceratops*, is essential to acquire information of neurological adaptation of ceratopsid dinosaurs. In this study, we obtained the CT scans of the braincases of *Triceratops* and made a quantitative comparison of the endocranial endocasts of them and other ceratopsians.

## MATERIALS & METHODS

### Specimens and CT scanning

The braincases of *Triceratops* (FPDM-V-9677 and FPDM-V-9775) were scanned using a micro-focus X-ray CT XT H 450 (Nikon) at Nikon Instech High-Resolution X-ray CT Facility, Nikon Instech Yokohama Plant, Kanagawa, Japan. These specimens are stored at Fukui Prefectural Dinosaur Museum (FPDM), Fukui, Japan. They were collected from the Hell Creek Formation (Upper Cretaceous), Ziebach, South Dakota for FPDM-V-9677, and Marmarth, North Dakota for FPDM-V-9775, U.S.A. FPDM-V-9677 consists not only of braincase but also of nasal, premaxillae, maxillae, rostrum, postorbital, jugal, quadrate, parietal, and squamosal (Figs. S1, S2). On the other hand, FPDM-V-9775 is an isolated braincase lacking other skull elements (Fig. S2). For FPDM-V-9677, CT images were acquired under a voltage of 445 kV, current of 460 µA, interslice spacing of 0.25 mm and image size of $1,947 \times 1,998$ pixels. For FPDM-V-9755, CT images were acquired under a voltage of 445 kV, current of 570 µA, interslice spacing of 0.25 mm and image size of $889 \times 1,267$ pixels. These parameters resulted in a voxel size of 0.25 mm along the $z$-axis and 0.20–0.25 mm in the $x$- and $y$-axes. The CT images revealed that FPDM-V-9775 is undeformed while its basicranial portion is missing.

We subsequently prepared the virtual endocasts of the cranial cavities and bony labyrinths from the acquired CT images using Amira (v 2019.3, Mercury Computer Systems, San Diego, CA, USA) (Figs. 1–3). Details of the methods used to prepare and examine the endocast models are provided by *Corfield et al. (2008)*. As is generally, the brain of reptiles, including most dinosaurs, does not fill the endocranium (*Jerison, 1973*; *Hopson, 1979*). Thus, endocasts do not provide complete information about brain morphology. Particularly the posterior part of the endocast tends to be larger than actual brain shape and size (*Watanabe et al., 2019*). Despite these limitations, endocasts still provide the best first-hand information on brain size and shape in extinct species.

### Skull size and body mass estimation of *Triceratops* specimens

To reconstruct total skull lengths for FPDM-V-9677 and 9775, we calculated the maximum cross-sectioned area for occipital condyles using their widths and heights (*Anderson, 1999*). *Anderson (1999)* demonstrated a correlation between the occipital condyle area and total skull length of *Triceratops* and obtained the following regression equation:

$$Y = 0.464\,X + 1.416,$$

where $Y$ = log total skull length (mm) and $X$ = log occipital condyle area (mm$^2$). The occipital condyle height and width of FPDM-V-9677 are 95 mm and 94 mm, respectively,

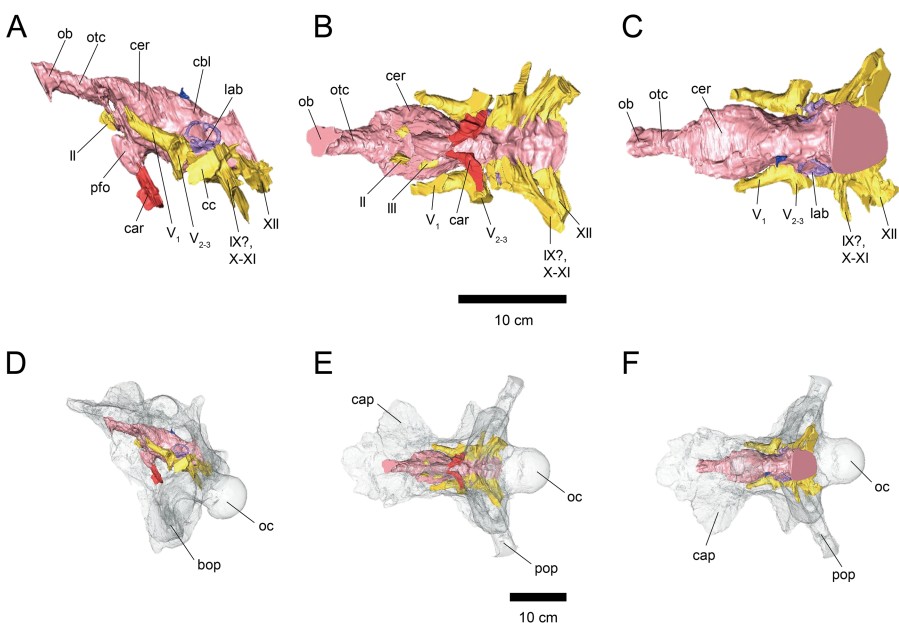

**Figure 1** **Cranial endocast (A–C) and braincase (D–F) of FPDM-V-9677.** (A, D) Left lateral, (B, E) ventral, and (C, F) dorsal views. Brain endocast represented by pink coloring; cranial nerves by yellow; carotid artery by red; venous canals by blue; endosseous labyrinth by purple. Abbreviations: car, cerebral carotid artery canal; c, cochlea; cc, columellar canal; cbl, cerebellum; cer, cerebral hemisphere; cvcm, caudal middle cerebral vein; lab, labyrinth; ob, olfactory bulb; otc, olfactory tract; II, optic nerve canal; III, oculomotor nerve canal; IV, trochlear nerve canal; $V_1$, ophthalmic nerve canal; $V_{2-3}$, maxillomandibular nerve canal; VI, abducens nerve canal; VII, facial nerve canal; IX-XI, shared canal for glossopharyngeal, vagus, and accessory nerves; XII, hypoglossal nerve canal; bop, basioccipital process; cap, capitate process of laterosphenoid; oc, occipital condyle; pop, paroccipital process.

resulting in the occipital condyle area of 7,014 mm². On the other hand, the occipital condyle height and width of FPDM-V-9775 are 87 mm and 91 mm, respectively, resulting in the occipital condyle area of 6,217 mm². Substituting these values of the occipital condyle areas for X yields the total skull lengths of 1,585 mm for FPDM-V-9677 and 1,514 mm for FPDM-V-9775. These estimates must be accepted as approximations, as the fit for the regression proposed by *Anderson (1999)* is not statistically significant ($p \sim 0.064$), and the sample size to derive the regression equation was small ($n = 5$).

The estimated total skull lengths of FPDM-V-9677 and 9775 are close to the measured total skull lengths of BSP1964 I458 (formerly YPM 1834) (*Ostrom & Wellnhofer, 1986*). Therefore the body mass estimate for BSP1964 I458 (4,963.6 kg) by *Seebacher (2001)* was used as the best body mass estimate for our specimens.

## Linear measurements

The maximum linear dimensions of the olfactory bulb region of FPDM-V-9677 were measured to estimate the degree of relative development of the olfactory bulb for *Triceratops*. We followed *Zelenitsky, Therrien & Kobayashi (2009)* and compared their value with that of other archosaurs. Ratios of the maximal linear dimensions of the olfactory bulb against that of cerebral hemisphere (log olfactory ratio) were calculated.

To assess the degree of development of semicircular canals in *Triceratops*, we calculated the ratio between the height and external diameter of the ASC, and the ratio between the total height of the PSC and height of PSC below the plane of lateral semicircular canal (LSC) for FPDM-V-9677 and 9775. We compared them by collecting such data via a literature search on other ceratopsians (*Brown, 1914*; *Hopson, 1979*; *Zhou et al., 2007*; *Witmer & Ridgely, 2008*), ankylosaurians (*Domínguez Alonzo et al., 2004*), pachycephalosaurians (*Domínguez Alonzo et al., 2004*; *Bourke et al., 2014*), ornithopods (*Domínguez Alonzo et al., 2004*; *Evans, Ridgely & Witmer, 2009*), sauropods (*Knoll et al., 2012*), and theropods (*Witmer & Ridgely, 2009*; *Azuma et al., 2016*), and adding these values to a plot presented in *Domínguez Alonzo et al. (2004)*.

The endosseous cochlear duct length (CL) of *Triceratops* was obtained from FPDM-V-9775. Additionally, those of other dinosaurs were measured from the figures in the literature (*Witmer & Ridgely, 2008*; *Evans, Ridgely & Witmer, 2009*; *Witmer & Ridgely, 2009*; *Leahey et al., 2015*; *Paulina-Carabajal, Lee & Jacobs, 2016*). The basilar papilla lengths of FPDM-V- 9775 and other dinosaurs with known CL were calculated following *Gleich, Dooling & Manley (2005)*. These basilar papilla lengths were assigned to a regression equation $Y = 5.7705 \, e^{-0.25X}$ ($X$ = basilar papilla length, $Y$ = best frequency of hearing) (*Gleich, Dooling & Manley, 2005*) to calculate the best frequency of hearing of *Triceratops*. The calculated best frequency of hearing was assigned to a regression equation, $Y = 1.8436 \, X + 1.0426$ ($X$ = best frequency of hearing, $Y$ = high frequency of hearing limit), to calculate the high frequency of hearing limit.

## RESULTS

The length from the rostral margin of the cerebrum to the caudal margin of the medulla of FPDM-V-9677 is 157 mm and its total length including the olfactory bulbs is 220 mm. The olfactory bulbs of FPDM-V-9677 are preserved, and the olfactory tracts extend ventrally (Fig. 1). The rostral margin of the olfactory bulbs is obscured. The endocast volume including that of olfactory bulbs is 434 cm$^3$. The volume of olfactory bulbs is 30 cm$^3$. In the forebrain region of FPDM-V-9775, olfactory bulb, olfactory tracts, and a portion of cerebral hemispheres are missing (Fig. 2). The rostrocaudal length of FPDM-V-9775 is 136 mm, and its total volume is 338 cm$^3$.

The cerebral hemispheres are dorsoventrally broad in FPDM-V-9677 (Fig. 1). The cerebrum of FPDM-V-9677 appears as a rounded swelling as in that of *Pachyrhinosaurus lakustai* (*Witmer & Ridgely, 2008*). Caudal to the cerebrum, the optic tectum is not clearly visible, and the cerebellum is indistinct from the hindbrain region in both specimens (Figs. 1 and 2). Additionally, there is no indication that the cerebellum had a floccular lobe. While the optic tectum of *Triceratops* has been suggested discriminable from the midbrain region (*Forster, 1996*), *Hopson (1979)* was unable to identify this structure. We also were unable to observe the optic tectum in our specimens.

Vascular elements may also be distinguished in the endocasts (Figs. 1 and 2). The caudal middle cerebral veins are visible in the endocast of FPDM-V-9775, and located between the paired semicircular canals and trigeminal nerves (Fig. 2). Although the pituitary region

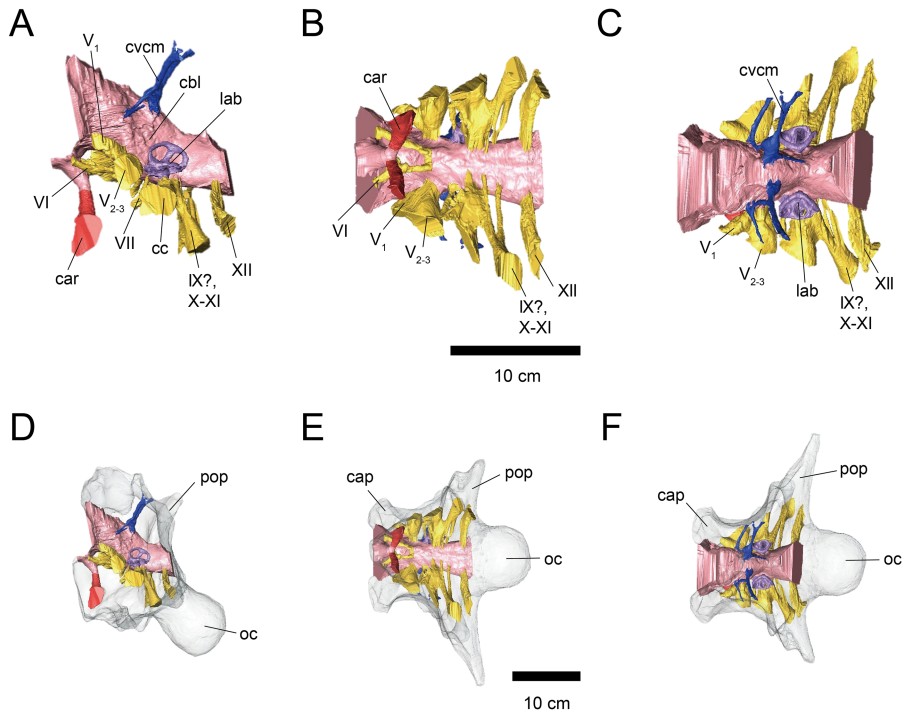

**Figure 2** **Cranial endocast (A–C) and braincase (D–F) of FPDM-V-9775.** (A, D) Left lateral, (B, E) ventral, and (C, F) dorsal views. Color scheme and abbreviations are given in the caption of Fig. 1.

is preserved in both specimens, its preservation in FPDM-V-9775 is only partial (Fig. 2). The pituitary region is funnel-shaped, and the carotid arteries extend from the ventral tip of this region in both specimens (Figs. 1 and 2).

## Cranial nerves

Endocasts do not represent illustrated traces of cranial nerves only and reflect the morphology of the cranial nerves and their accompanying assemblage of soft tissues, such as blood vessels. However, since it is difficult to observe cranial nerves and other soft tissues in isolation, we will focus mainly on cranial nerves here.

The olfactory system is not preserved in FPDM-V-9775, while it can be identified in FPDM-V-9677 (Fig. 1). The olfactory tracts extend from the cerebral hemisphere and are relatively long. The thickness of the olfactory bulbs varies among specimens, and FPDM-V-9677 appears to be relatively thin. The olfactory bulbs of SMM P 2014.3.1C (*Erickson, 2017*) are much thinner and more dorsally inclined than in FPDM-V-9677 and other specimens (e.g., *Bürckhardt, 1892*; *Marsh, 1896*; *Hay, 1909*; *Hopson, 1979*; *Forster, 1996*). Those striking morphological differences of the olfactory bulbs may be due to a molding artifact, or that the olfactory bulbs are difficult to observe as an endocast in the archosaurs in general.

Both the right optic nerve (CN II) canal and the oculomotor nerve (CN III) canal are preserved in FPDM-V-9677 (Fig. 1). The trochlear nerve (CN IV) canal cannot be reconstructed in either of the two specimens because of the preservation.
The trigeminal nerve (CN V) is located at the rostral end of the medulla, rostrally adjacent to CN VII in both specimens (Figs. 1 and 2). The ophthalmic nerve (CN $V_1$) canal extends rostrally to the middle level of the cerebrum in FPDM-V-9677, and the maxillomandibular nerve (CN $V_{2-3}$) canal extends laterally at a right angle to the ophthalmic branch (Fig. 1). Although CN $V_1$ and CN $V_{2-3}$ follow the same course outward from the brain, they run separated through the braincase and, thus, appear from different foramina, as observed in other chasmosaurines such as *Anchiceratops* (*Hopson, 1979*). This is in contrast to centrosaurines like *Pachyrhinosaurus*, which shows the two trigeminal nerve trunks branching from the endocast in closer association (*Witmer & Ridgely, 2008*; *Tykoski & Fiorillo, 2013*).

The abducens nerve (CN VI) canal is preserved only in FPDM-V-9775 and passes rostroventrally from the rostroventral end of the medulla through the both sides of the pituitary (Fig. 2). The facial nerve (CN VII) canal is visible on both sides of FPDM-V-9775. The vestibulocochlear nerve (CN VIII) canal is not preserved in either of the two specimens.

The glossopharyngeal nerve (CN IX), vagus nerve (CN X), and accessory (CN XI) nerve may be linked together, extending caudolaterally from the lateral region of the medulla (Figs. 1 and 2). These nerves exit the vagal canal located posterior to the inner ear. *Erickson (2017)* and other previous studies (e.g., *Bürckhardt, 1892*; *Marsh, 1896*; *Hay, 1909*; *Gilmore, 1919*; *Hopson, 1979*; *Forster, 1996*) seem to have misidentified columellar canals as the canal of CN VII, VIII, or IX-XI. The hypoglossal nerve (CN XII) canal can be observed in both specimens (Figs. 1 and 2), passing caudolaterally to exit through one opening located in the exoccipital, as seen in *Pachyrhinosaurus lakustai* (*Witmer & Ridgely, 2008*).

### Endosseous labyrinth

The labyrinths of the inner ears are preserved on both sides of the specimens under study. The left inner ear of FPDM-V-9775 is particularly well preserved (Figs. 2 and 3). The semicircular canals are located caudolaterally to the cerebellum in both FPDM-V-9677 and 9775 (Figs. 1 and 2). The ASC is round in general morphology. In particular, the arc of the ASC is relatively low dorsoventrally, differing from tall arcs of both *Psittacosaurus* (*Zhou et al., 2007*; *Bullar et al., 2019*; *Napoli et al., 2019*) and *Protoceratops* (*Hopson, 1979*). The PSC is slightly lower dorsoventrally than the ASC when the LSC is oriented horizontally (Fig. 3). The LSC is the shortest in length of the three canals. The cochlear duct is preserved in FPDM-V-9775 ventral to the vestibular apparatus (Fig. 3), and its length is 17.95 mm, longer than those of the lambeosaurine hadrosaurids (*Evans, Ridgely & Witmer, 2009*) and of *Pachyrhinosaurus* (*Witmer & Ridgely, 2008*). The ratio between the height and external diameter of the ASC (relative height of the ASC) of FPDM-V-9775 is 0.85, whereas the ratio between the height of the PSC and the height of the PSC below the plane of LSC (relative degree of ventral expansion of the PSC below the plane of LSC) is 0.29.

## DISCUSSION

### Skull size and body weight

We estimated the total skull length of both FPDM-V-9677 and FPDM-V-9775 from occipital condyle area (*Anderson, 1999*), resulting in roughly the same skull length between

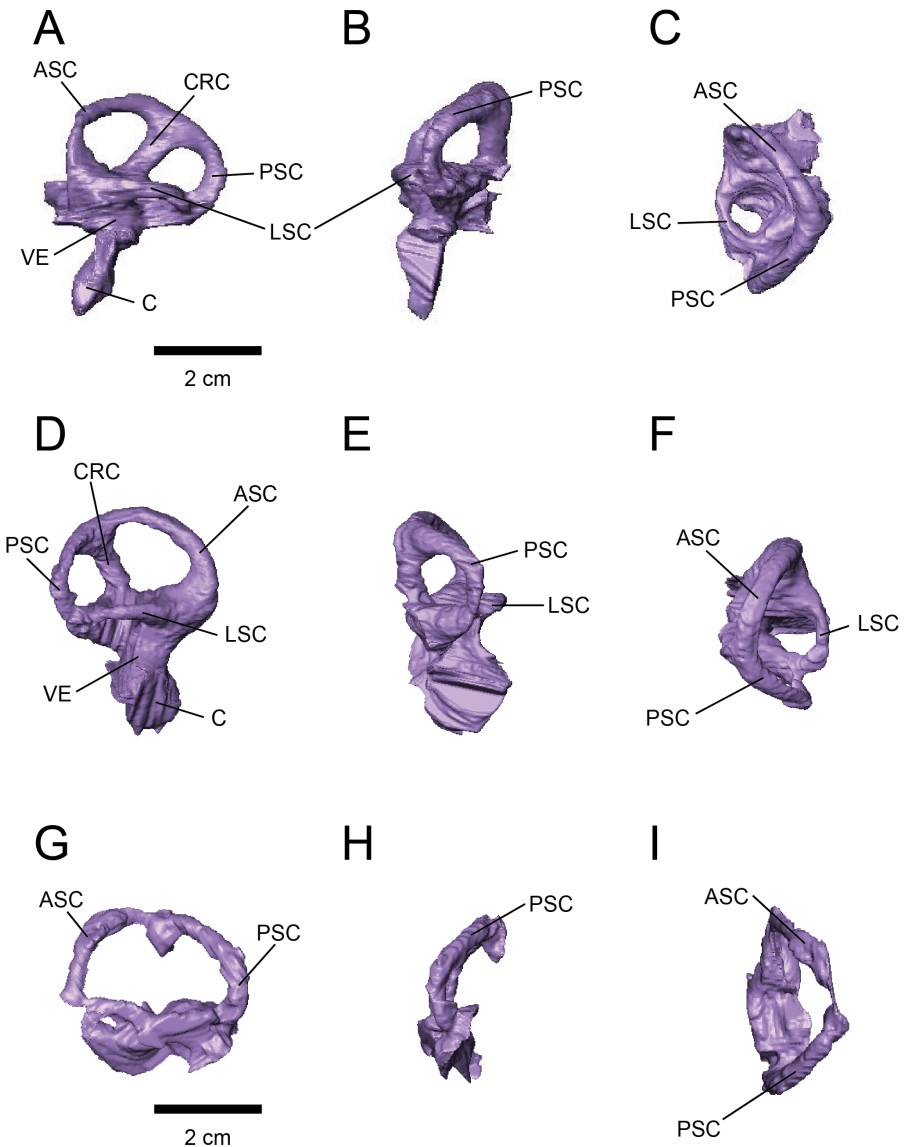

**Figure 3   Left (A–C) and right (D–H) endosseous labyrinth of FPDM-V-9775, and left (G–H) endosseous labyrinth of FPDM-V-9677.** (A, D, G) Lateral, (B, E, H) posterior, and (C, F, I) dorsal views. Abbreviations: ASC, anterior semicircular canal; C, cochlea; CRC, crus commune; LSC, lateral semicircular canal; PSC, posterior semicircular canal; VE, vestibule of inner ear.

the two specimens: 159.4 cm and 151.1 cm, respectively. Therefore, we assume both *Triceratops* specimens are of approximately the same body weight.

## Olfactory bulbs and sense of smell

Olfactory bulb size has been used as an indicator of the acuity of the sense of smell in extant mammals and archosaurs, and a positive correlation has been reported between the olfactory bulb size and olfactory acuity (*Cobb, 1960*; *Zelenitsky, Therrien & Kobayashi, 2009*). We calculated the olfactory ratios of FPDM-V-9677 (*Triceratops*), *Corythosaurus* sp.,

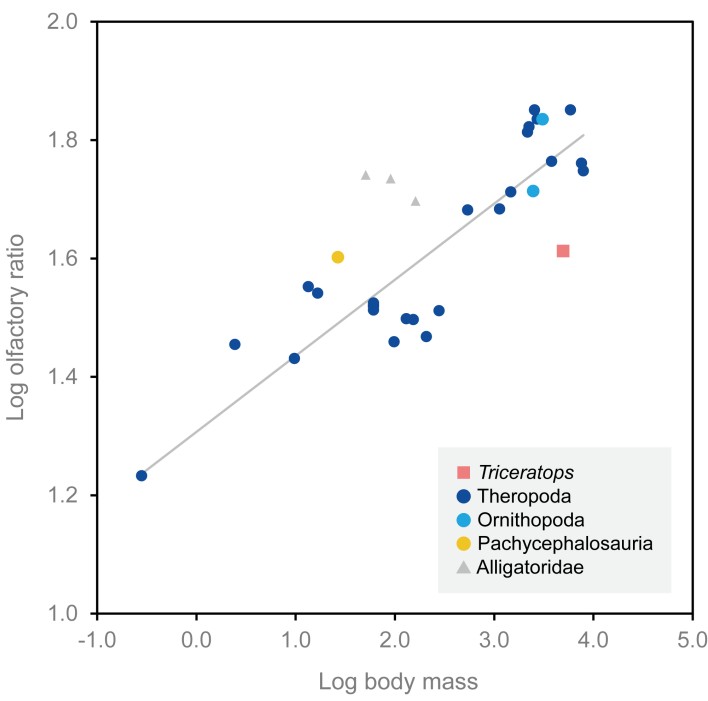

**Figure 4   Relationship between olfactory ratio and body mass for selected dinosaurs.** After *Zelenitsky, Therrien & Kobayashi (2009)*. See Table 1 for additional data for *Triceratops*, *Corythosaurus*, *Hypacrosaurus* and *Stegoceras*.

*Hypacrosaurus altispinus* (*Evans, Ridgely & Witmer, 2009*), and *Stegoceras validum* (*Bourke et al., 2014*) following *Zelenitsky, Therrien & Kobayashi (2009)* and compared these data in a scatter plot for relative olfactory bulb size against body mass for theropods and other archosaurs (Fig. 4; Table 1). Consequently, *Triceratops* is plotted considerably below the regression line, i.e., olfactory ratio to body mass of other dinosaurs, indicating that the acuity of the sense of smell of *Triceratops* was lower than the average of other dinosaurs and alligators (Fig. 4). Our result contrasts with the observations from *Psittacosaurus* (*Zhou et al., 2007*), a primitive ceratopsian, which had enlarged olfactory bulbs, but is concordant with the small size of the olfactory bulbs in *Pachyrhinosaurus* (*Witmer & Ridgely, 2008*). Due to the lack of available body mass data for other ceratopsians, it is impossible to calculate their olfactory ratios for quantitative analysis. Nonetheless, it is hypothesized that ceratopsians reduced their sense of smell in the course of evolution.

## Alert head posture

The alert head posture of extinct animals can be evaluated by orienting the LSC horizontally (*Duijm, 1951*; *Witmer & Ridgely, 2008*). Although the orientation and morphology of the LSC are variable intra- and interspecifically, the LSC is useful for reconstructing head posture and locomotion of extinct animals, taking into account the degree of morphological variation and phylogeny (*Duijm, 1951*; *Marugán-Lobón, Chiappe & Farke, 2013*; *Berlin, Kirk & Rowe, 2013*; *Coutier et al., 2017*). In fact, the estimation of head posture using LSC

**Table 1  Olfactory ratios and body masses of *Triceratops, Corythosaurus, Hypacrosaurus* and *Stegoceras*.** Olfactory ratios calculated following the method of *Zelenitsky, Therrien & Kobayashi (2009)*.

| Species | Specimen number | Olfactory ratio (%) | Log olfactory ratio | Body mass (kg) | Log body mass | Reference (brain endocast) | Reference (body mass) |
|---|---|---|---|---|---|---|---|
| *Triceratops* sp. | FPDM-V-9677 | 40.98 | 1.613 | 4,963.6 | 3.696 | – | *Seebacher (2001)* |
| *Corythosaurus* sp. | CMN 34825 | 68.42 | 1.835 | 3,078.5 | 3.488 | *Evans, Ridgely & Witmer (2009)* | *Evans, Ridgely & Witmer (2009)* |
| *Hypacrosaurus altispinus* | ROM 702 | 51.72 | 1.714 | 2,478.0 | 3.394 | *Evans, Ridgely & Witmer (2009)* | *Evans, Ridgely & Witmer (2009)* |
| *Stegoceras validum* | UALVP 2 | 40.00 | 1.602 | 26.7 | 1.427 | *Bourke et al. (2014)* | *Seebacher (2001)* |

has been conducted in the ceratopsian *Anchiceratops* (*Tait & Brown, 1928*). On the other hand, a study using the Procrustes method have concluded that prediction of alert head posture by LSC is difficult since variability of LSC relative to skull landmarks of dinosaurs are large (*Marugán-Lobón, Chiappe & Farke, 2013*). Therefore, estimated alert posture of the head by LSC should be accepted with caution.

By adjusting the braincase of FPDM-V-9775 in a reconstructed skull model, it was found that the beak is oriented relatively downward (Fig. 5). Thus, *Triceratops* likely had an alert head posture such that the basicranial axis was inclined approximately 45° below the horizontal plane. At this head posture, their two horns and frill would have faced straight forward at an angle efficient for displaying, and their beak inclined slightly to the ground to facilitate grazing. *Ostrom & Wellnhofer (1986)* found that when the inferior margin of the maxilla of *Triceratops* is horizontal, the occipital condyle projects downward by about 30 to 35 degrees, indicating that the head was carried in a "pitch forward" posture, which is supported by the alert head posture based on LSC in this study.

## Stabilization of gaze and posture

Semicircular canals are related to the sense of balance, equilibrium, agility of locomotion, and stabilization of gaze (*Spoor et al., 2007*; *Witmer et al., 2008*). To assess the developmental degree of semicircular canals, *Domínguez Alonzo et al. (2004)* calculated the ratio between the height and external diameter of the ASC and the ratio between the height of the PSC and the PSC below the plane of LSC of *Archaeopteryx* and compared this to that of Aves, non-archosaur reptiles and archosaurs. We calculated these ratios for *Triceratops* (FPDM-V-9677 and 9775) and compared them to those of other dinosaurs (Fig. 6, Table 2). Although *Domínguez Alonzo et al. (2004)* used a large set of data for extant and extinct archosaurs, it seems not the case for non-avian dinosaurs. Therefore, we included additional data for the animals based on literature published after *Domínguez Alonzo et al. (2004)*. According to the scatter plot, *Triceratops* and other derived ceratopsians (*Anchiceratops* and *Pachyrhinosaurus*) are plotted in a lower area than *Psittacosaurus* and *Protoceratops* (Fig. 6), indicating that the vertical semicircular canals (ASC and PSC) of *Triceratops* and other derived ceratopsians are less well-developed than those of primitive ceratopsians (*Psittacosaurus* and *Protoceratops*). Thus, we conclude that sensory input for the reflexive stabilization of gaze and posture in *Triceratops* was lower than those of primitive ceratopsians. The ASC is also strongly correlated with locomotor mode. It has been suggested that bipedal dinosaurs exhibit well-developed ASC, while quadrupedal dinosaurs do not (*Georgi, Sipla & Forster, 2013*). Therefore, our observation that primitive, bipedal ceratopsians have better-developed ASC than derived, quadrupedal ceratopsians may reflect the difference in their locomotor modes. It should be noted that plotting ratios against ratios does not take into account the effects of allometry and may hinder information and variability in the data. However, the ASC and the ventral part of the PSC are prominent in birds, most of which exhibit exceptional three-dimensional motility among terrestrial vertebrates. Thus, the development of ASC and the ventral part of the PSC are very likely correlated with animal motility. In addition, the strong reduction of the LSC is observed in quadrupedal, less-mobile sauropods (*Witmer et al., 2008*) and this

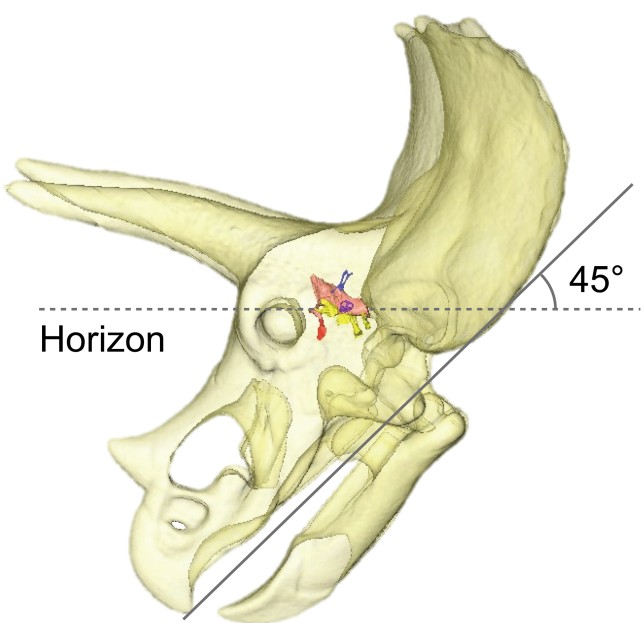

**Figure 5** Alert head posture of *Triceratops* based on orienting the skull such that the lateral semicircular canal is horizontal.

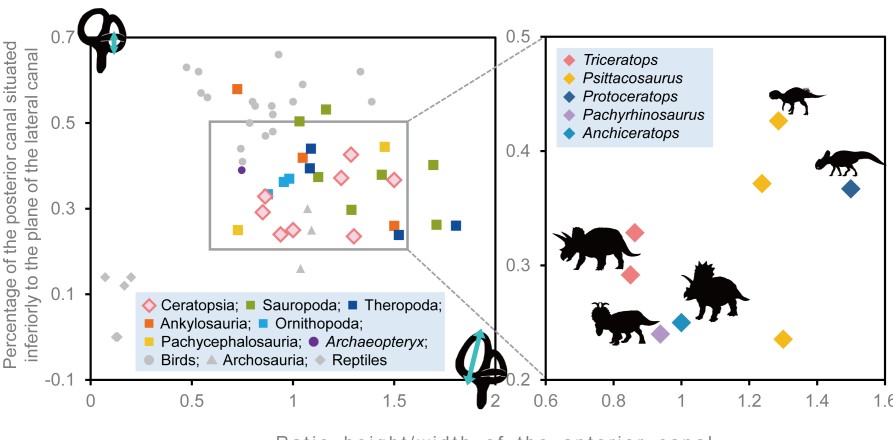

**Figure 6** Comparative proportions of the inner ears of ceratopsians, selected recent birds, archosaurs and non-archosaur reptiles.

configuration is similar to that of *Triceratops* in this study. *Witmer et al. (2008)* suggested that mediolateral eye and head movements were less important to sauropods because their LSC is short. Similarly, *Triceratops* probably was not well adapted to the mediolateral head movement.

**Table 2 Proportions of the inner ear of *Triceratops* and selected dinosaurs.** "Relative anterior canal height" represents the height of the anterior canal/external diameter of anterior canal. "Relative posterior canal height" represents the height from the base of the posterior canal to the plane of the lateral canal/height of the posterior canal.

| Species | Specimen number | Relative anterior canal height | Relative posterior canal height | Reference |
|---|---|---|---|---|
| *Triceratops* sp. | FPDM-V-9677 | 0.863 | 0.328 | – |
| *Triceratops* sp. | FPDM-V-9775 | 0.85 | 0.292 | – |
| *Psittacosaurus lujiatunensis* | IVPP V15451 | 1.286 | 0.426 | *Bullar et al. (2019)* |
| *Psittacosaurus lujiatunensis* | IVPP V12617 | 1.237 | 0.371 | *Bullar et al. (2019)* |
| *Psittacosaurus lujiatunensis* | PKUP V1054 | 1.300 | 0.235 | *Zhou et al. (2007)* |
| *Protoceratops grangeri* | AMNH 6466 | 1.500 | 0.367 | *Hopson (1979)* |
| *Pachyrhinosaurus lakustai* | TMP 1989.55.1243 | 0.938 | 0.24 | *Seebacher (2001)* and *Witmer & Ridgely (2008)* |
| *Anchiceratops ornatus* | AMNH 5259 | 1 | 0.25 | *Brown (1914)* |
| *Hypacrosaurus altispinus* | ROM 702 | 0.876 | 0.334 | *Evans, Ridgely & Witmer (2009)* |
| *Corythosaurus* sp. | CMN 34825 | 0.954 | 0.363 | *Evans, Ridgely & Witmer (2009)* |
| *Iguanodon bernissartensis* | BMNH R 2501 | 0.981 | 0.37 | *Domínguez Alonzo et al. (2004)* |
| *Euoplocephalus tutus* | AMNH 5337 | 1.5 | 0.26 | *Leahey et al. (2015)* |
| *Kunbarrasaurus ieversi* | QM F18101 | 0.724 | 0.577 | *Leahey et al. (2015)* |
| *Pawpawsaurus campbelli* | FWMSH93B.00026 | 1.048 | 0.417 | *Paulina-Carabajal, Lee & Jacobs (2016)* |
| *Pachycephalosaurus grangeri* | AMNH 1696 | 0.727 | 0.25 | *Domínguez Alonzo et al. (2004)* |
| *Stegoceras validum* | UALVP 2 | 1.452 | 0.444 | *Bourke et al. (2014)* |
| *Massospondylus carinatus* | BP/1/4779 | 1.289 | 0.297 | *Knoll et al. (2012)* |
| *Spinophorosaurus nigerensis* | GCP-CV-4229 | 1.709 | 0.262 | *Knoll et al. (2012)* |
| *Nigersaurus taqueti* | MNN GAD512 | 1.163 | 0.532 | *Knoll et al. (2012)* |
| *Diplodocus longus* | CM 3452 | 1.693 | 0.402 | *Knoll et al. (2012)* |
| *Camarasaurus lentus* | CM 11338 | 1.125 | 0.374 | *Knoll et al. (2012)* |
| *Giraffatitan brancai* | MB.R.2180.22.1-4 | 1.439 | 0.379 | *Knoll et al. (2012)* |
| *Jainosaurus septentrionalis* | ISI R162 | 1.032 | 0.504 | *Knoll et al. (2012)* |
| *Tyrannosaurus rex* | AMNH FR 5117 | 1.083 | 0.394 | *Witmer & Ridgely (2009)* |
| *Gorgosaurus libratus* | ROM 1247 | 1.523 | 0.239 | *Witmer & Ridgely (2009)* |
| *Allosaurus fragilis* | UMNH VP 18050 | 1.804 | 0.260 | *Witmer & Ridgely (2009)* |
| *Fukuivenator paradoxus* | FPDM-V8461 | 1.089 | 0.44 | *Azuma et al. (2016)* |

## Hearing ability

In FPDM-V-9775, the CL and basilar papilla length are longer than those of other dinosaurs analyzed in this study with exceptions of those of *Pawpawsaurus* and *Kunbarasaurus* (Fig. 7; Table 3). A basilar papilla length is defined by *Gleich, Dooling & Manley (2005)* as two-thirds of the corresponding CL. Although *Evans, Ridgely & Witmer (2009)* calculated the best frequency of hearing for Lambeosaurines using the equation derived in *Gleich, Dooling & Manley (2005)*, their calculation was based on the CL rather than the basilar papilla length as originally proposed by *Gleich, Dooling & Manley (2005)*. Therefore, the calculated hearing frequencies in this study do not match those of *Evans, Ridgely & Witmer (2009)*.

Following *Gleich, Dooling & Manley (2005)*, the best frequency of hearing for *Triceratops* is estimated 290 Hz based on the basilar papilla length of FPDM-V-9677 and 9775. Although

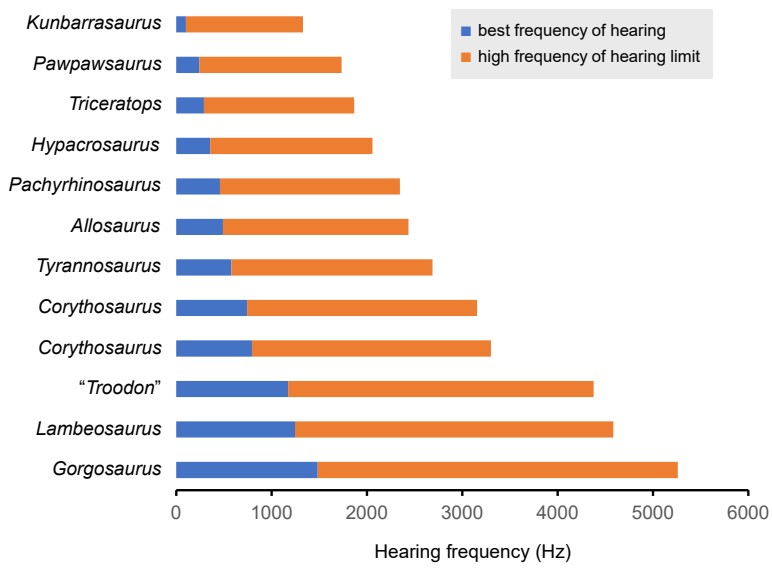

**Figure 7** **Best frequency of hearing and high frequency of hearing limit for selected dinosaurs calculated from the regression of** *Gleich, Dooling & Manley (2005)*. Blue color shows best frequency of hearing (Hz), and orange color shows high frequency of hearing limit (Hz).

**Table 3** **Values of cochlear lengths (CL), calculated basilar papilla length, best frequency of hearing and high frequency of hearing limit.** "Basilar papilla length", "Best frequency of hearing" and "High frequency of hearing limit" were calculated following the method of *Gleich, Dooling & Manley (2005)*.

| Species | Specimen number | CL (mm) | Basilar papilla length (mm) | Best frequency of hearing (Hz) | High frequency of hearing limit (Hz) | Reference |
|---|---|---|---|---|---|---|
| *Triceratops* sp. | FPDM-V-9775 | 17.95 | 12.0 | 0.290 | 1.577 | – |
| *Pachyrhinosaurus lakustai* | TMP 1989.55.1243 | 15.2 | 10.1 | 0.458 | 1.887 | *Witmer & Ridgely (2008)* |
| *Tyrannosaurus rex* | AMNH FR 5117 | 13.8 | 9.2 | 0.579 | 2.109 | *Witmer & Ridgely (2009)* |
| *Gorgosaurus libratus* | ROM 1247 | 8.15 | 5.4 | 1.484 | 3.778 | *Witmer & Ridgely (2009)* |
| *Allosaurus fragilis* | UMNH VP 18050 | 14.8 | 9.9 | 0.490 | 1.945 | *Witmer & Ridgely (2009)* |
| "*Troodon formosus*" | composite of TMP 86.36.457 and TMP 79.8.1 | 9.56 | 6.4 | 1.173 | 3.205 | *Witmer & Ridgely (2009)* |
| *Hypacrosaurus altispinus* | ROM 702 | 16.7 | 11.1 | 0.357 | 1.700 | *Evans, Ridgely & Witmer (2009)* |
| *Lambeosaurus* sp. | ROM 758 | 9.2 | 6.1 | 1.245 | 3.339 | *Evans, Ridgely & Witmer (2009)* |
| *Corythosaurus* sp. | ROM 759 | 11.9 | 7.9 | 0.794 | 2.507 | *Evans, Ridgely & Witmer (2009)* |
| *Corythosaurus* sp. | CMN 34825 | 12.3 | 8.2 | 0.743 | 2.412 | *Evans, Ridgely & Witmer (2009)* |
| *Kunbarrasaurus ieversi* | QM F18101 | 24.3 | 16.2 | 0.101 | 1.228 | *Leahey et al. (2015)* |
| *Pawpawsaurus campbelli* | FWMSH93B.00026 | 19 | 12.7 | 0.243 | 1.491 | *Paulina-Carabajal, Lee & Jacobs (2016)* |

the hearing ranges of multiple dinosaur taxa were calculated in this study, these are outside the range of the original data used to derive the equation in *Gleich, Dooling & Manley (2005)*. Therefore, it was necessary to extrapolate the regression and the confidence interval of *Gleich, Dooling & Manley (2005)* to assess the hearing ranges of dinosaurs, accepting

that the extrapolation may result in overestimation or underestimation of the true hearing ranges of the animals. Nonetheless, compared to other dinosaurs, *Triceratops* appears to have been adapted to hearing relatively lower frequency. As low frequencies would be less susceptible to scattering and reflection by objects in the path of the sound (*Poole et al., 1988*; *Lewis & Fay, 2004*), *Triceratops* may have been sensitive to sounds from long distances.

## CONCLUSIONS

Based on our interpretations of the endocranial anatomy of *Triceratops*, we suggest that (1) the sense of smell was lower than those of most, if not all, other dinosaurs; (2) the alert head posture was angled so that their frills and horns faced front against potential threads, and their beaks pointed to the ground to facilitate grazing; (3) mean hearing frequency was relatively lower among dinosaurs; and (4) lower ability to stabilize gaze inhibited rapid head movements compared to primitive ceratopsians, such as *Psittacosaurus* and *Protoceratops*.

## ACKNOWLEDGEMENTS

We thank the staffs at Nikon Instech's High-Resolution X-ray CT Facility for access and assistance to conduct CT analyses, and the researchers and staffs of Fukui Prefectural Dinosaur Museum for providing information about the specimens. The members of the Institute of Dinosaur Research, Fukui Prefectural University provided helpful comments and suggestions for this study. This manuscript was greatly improved by comments from Brandon Hedrick (Louisiana State University Health Sciences Center), Andrew Farke (Raymond M. Alf Museum of Paleontology), Jason Bourke (New York Institute of Technology College of Osteopathic Medicine) and an anonymous reviewer. The authors would also like to thank Enago for the English language review.

### Funding

The authors received no funding for this work.

### Competing Interests

The authors declare there are no competing interests.

### Author Contributions

- Rina Sakagami conceived and designed the experiments, performed the experiments, analyzed the data, prepared figures and/or tables, authored or reviewed drafts of the paper, and approved the final draft.
- Soichiro Kawabe conceived and designed the experiments, performed the experiments, analyzed the data, authored or reviewed drafts of the paper, and approved the final draft.

### Data Availability

Raw data is available in Tables 1–3.

Both specimens in this study (specimen numbers: FPDM-V-9677 and FPDM-V-9775) are stored at Fukui Prefectural Dinosaur Museum.

## Supplemental Information

Supplemental information for this article can be found online at http://dx.doi.org/10.7717/peerj.9888#supplemental-information.

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
