# Peer review of "Endocranial anatomy of the ceratopsid dinosaur Triceratops and interpretations of sensory and motor function"

_PeerJ, doi:10.7717/peerj.9888_

## Round 0.1 · original submission · Major Revisions

Thank you for your submission. I agree with the reviewers that the paper is a valuable contribution and is especially welcome as it incorporating quantitative metrics into endocast studies.

However, there are a number of revisions that should be made. In general, I think you should be more careful about some of your broad interpretive statements. There are limitations inherent in many of your analyses that are not clearly expressed. Particularly, as mentioned by the reviewers, Hertz values cannot be negative, and this is likely a result of applying an equation derived for small animals to Triceratops. Additionally, I would like to see more comparisons with previous literature in your qualitative section. For example, Forster (1996) described the endocranial anatomy of Triceratops, but this paper is not compared in detail with your interpretations. There is a new paper on the endocranial anatomy of Auroraceratops that should be included as well (Zhang et al., 2019).

I think that this will be a strong paper once the corrections suggested by the reviewers are made. Let me know if you have any questions going forward and I look forward to your next draft.

·

Basic reporting

See general author comments.

Experimental design

See general author comments.

Validity of the findings

See general author comments.

Additional comments

This manuscript presents new data on the cranial endocast for the horned dinosaur Triceratops, based on two previously undescribed specimens. The basic anatomical data are a worthy contribution to the literature, and the anatomical identifications seem accurate based on the provided information. The functional interpretations and numerical analysis need considerable revision, through a deeper engagement with the literature in this area as well as more cautious interpretation of the raw results. For instance, calculations of best hearing frequency for non-avian dinosaurs by necessity must be based upon equations that were often calculated for much smaller organisms. Thus, the size of something like Triceratops may be far outside the range of where estimates can be considered reliable. This is, in part, why the results presented here show negative frequencies for some estimates (e.g., Table 3), which is not physically possible. Similarly, allometric effects may make it hard to interpret things like canal proportions, especially when comparing animals of very disparate size. I suggest a deep revision of these parts of the paper in particular, either removing them altogether, or a major reworking to limit and update the assumptions underlying the various functional analyses. Specific comments in these areas are outlined below.

DETAILED COMMENTS
- I would suggest rewording the title to remove the colon, and make the title more direct -- e.g., something like "Endocranial anatomy of Triceratops (Ceratopsidae: Dinosauria) and interpretations of sensory and motor function" or "Endocranial anatomy of the ceratopsid dinosaur Triceratops and interpretations of sensory and motor function" (totally optional, of course!)
- line 78: Forster 1996 should be added here.
- line 87-105: Can you provide more information on the thresholding protocol used? I suspect a fair amount of manual segmentation was required, but it would be helpful to note this. Maybe provide an example slice to show the general character of the scans?
- line 88: Some additional basic information shold be provided. How were these identified as Triceratops (versus Torosaurus)? Do you have enough of the skull to infer species (T. horridus vs. T. prorsus). Do you know where in section the fossils were recovered? Are the braincases crushed or distorted? Can you provide some basic dimensions so that size can be compared with other braincases and perhaps more precisely infer skull size? Optionally, a photo of the braincase would be helpful.
- line 115: was area calculated from width and height of the occipital condyle? If so, these should be provided as measurements, to be more directly comparable with other specimens. (area is not a common metric outside of the Anderson 1999 paper)
- line 122 (and relevant places later in description): the linear measurements, not just ratios, should be provided. Ratios are very hard to interpret, especially when allometric effects might be a factor. Some other studies have found allometric patterns in semicircular canal proportions, so I would use ratios with caution for any inferences. Departures from allometric expectations are of interest. Furthermore, the "raw" measurements are more easily used by future researchers for comparison with other specimens.
- line 148-154: Can you provide volume of the endocast for FPDM-V-9677 w/o the olfactory bulb? I would make sure to provide all comparable measurements possible relative to, e.g., P. lakustai
- lines 170-197: It is important to note that the various nerve canals didn't *just* carry nerves, but also had vasculature and other tissues. I might suggest a brief statement in the text to clarify this point--I am find with using the anatomical shorthand for simplicity, but do think an acknowledgment that these are only approximate reconstructions based on the bony dimensions, not a true representation of the exact form/size of the neural tissue.
- lines 199-212: It would be helpful to add some discussion on similiarites/differences between the two specimens. Is there any evidence of asymmetry in the individuals? I would suggest adding views to Figure 3, to show the right side of FPDM-V-9775, as well as views of the canals in FPDM-V-9677
- lines 256-276: I would suggest augmenting this section to include other discussions of the functional morphology of semicircular canals. The Georgi et al. 2013 paper is one. Coutier et al. 2017 and Berlin et al. 2013 also provide an overview of the use of the lateral canal in inferring head posture, along with some appropriate cautions.
- lines 285-292: This seems like a pretty big stretch of the evidence.
- It's interesting to note that Figure 5 in this paper compares very favorably with the head orientation hypothesized by Ostrom & Wellnhofer (1986), as shown in their Figure 7. See also Tait & Brown 1928, which uses semicircular canal orientation and other lines of evidence to show a similar posture for Anchiceratops.
- What is the source of the skull in Figure 5? How was the braincase orientation within the skull determined?
- Figure 6 is plotting a ratio against a ratio...this isn't necessarily wrong, but I do wonder if it isn't obscuring some information or variation within the data. What does this plot mean biologically? At the appropriate part in the text (lines 257-273), you might add some explanation of what these values are indicating biologically.
- The 2017 paper by Bruce Erickson (History of the Ceratopsian DInosaur Triceratops in the Science Museum of Minnesota) illustrates a Triceratops endocast and should be cited / referenced. Available here: https://www.smm.org/sites/default/files/public/attachments/history-ceratopsian-triceratops.pdf
- Brown & Schlaikjer 1940 provide some of the best illustrations and such of the endocast for Protoceratops, and so this would be useful to reference here.
- Can you comment on differences between these endocasts and others described for Triceratops (e.g., USNM 5740)? On USNM 5740, it looks like the olfactory region is more angled relative to the rest of the endocast. Is this distortion? Individual variation? Morhardt and colleagues (2018) had an abstract on Triceratops endocranial anatomy; it might be cited here (although it is only an abstract, so data are limited).
- I would strongly suggest that the CT scan data should be deposited in an appropriate digital repository. I recognize that museum policies may vary in terms of how freely this should be done (and it thus may have to require permission from a collections manager prior to granting access for other workers), but I would strongly advise this to allow greater reproducibility of results. Particularly for small structures, a difference of pixel interpretation or thresholding may alter measurements between observers.
- Raw data underlying the charts as well as the ratios should be provided in a supplemental table.
- Table 3: The calculation of negative values does not make sense (as far as I know, Hertz can only be a positive value). This is the fault of the original equation, which was only developed with a set of organisms much, much smaller than many non-avian dinosaurs (including Triceratops). Thus, I think the dinosaurs are far outside the limit of where the equation can be considered reliable. I would suggest removing this section from the paper, or else choosing a different method. Gleich et al. 2005 provide another equation, although this too suffers from the limitations of being calculated from animals much smaller than Triceratops.
- The Zhang et al. 2019 paper on Auroraceratops endocasts should probably be incorporated into the paper.

CITATIONS
Berlin JC, Kirk EC, Rowe TB. 2013. Functional Implications of Ubiquitous Semicircular Canal Non-Orthogonality in Mammals. PloS One 8:e79585

Coutier, F., Hautier, L., Cornette, R., Amson, E. and Billet, G., 2017. Orientation of the lateral semicircular canal in Xenarthra and its links with head posture and phylogeny. Journal of Morphology, 278(5), pp.704-717.

Georgi JA, Sipla JS, Forster CA (2013) Turning Semicircular Canal Function on Its Head: Dinosaurs and a Novel Vestibular Analysis. PLoS ONE 8(3): e58517. https://doi.org/10.1371/journal.pone.0058517

Gleich O, Dooling RJ, Manley G. 2005. Audiogram, body mass, and basilar papilla length: correlations in birds and predictions for extinct archosaurs. Naturwissenschaften 92:595–598.

Mordhardt, AC., et al. 2018. STUDY OF ENDOCRANIAL ANATOMY AND ONTOGENY IN THE LATE CRETACEOUS NON-AVIAN DINOSAUR GENUS TRICERATOPS. Journal of Vertebrate Paleontology, Program and Abstracts, 2018, p. .186

Zhang, Qian-Nan, et al. "Endocranial morphology of Auroraceratops sp.(Dinosauria: Ceratopsia) from the Early Cretaceous of Gansu Province, China." Historical Biology (2019): 1-6.

·

Basic reporting

Language is clear and unambiguous for the most part. I've listed some spelling and grammatical issues that should be addressed but these are mostly a lot of little things.

The largest issue here is in Figure 7.

The caption says that this is the mean best hearing, but the X axis only shows the scaled and transformed endosseous cochlear length. There is no Y axis showing hearing frequency here. I would change out the graph for a XY graph showing the hearing frequency or swap it out for just a histogram showing scaled cochlear lengths in the various dinosaurs.

The rest of these are a list of minor requested changes

Line 49: Change "it is considered" to: "it was inferred".

Lines 53–54: Delete "thus". Move the rest of the sentence up to line 44 right after
"Napoli et al. 2019)".

Line 55, 69, and elsewhere: Delete the extra modifier "ceratopsids". You can just say "centrosaurines" or "chasmosaurines". The entire article is about ceratopsians and dinosaurs in general. The audience will know what groups you are referring to.

Line 59: change "Pachyrhinosaurus has" to "Pachyrhinosaurus had".

Line 60: Delete "Because of the" and start the sentence at "Elongate".

Line 61. End the sentence after "(Spoor et al. 2007)". Start next sentence with: "The authors".

Lines 63–64. Sentence:

"In addition, because Pachyrhinosaurus has quite a short cochlear duct, it was suggested that"

Change to:

"In addition, Pachyrhinosaurus had a short cochlear duct, suggesting that"

Line 70: change "can" to: "could".

Line 74: change "is" to "was".

Line 81: Delete "thus".

Line 83: Delete "dinosaurs". Can simply end with "ceratopsians".

Lines 100–101: The section that reads:

"including non-theropod dinosaurs"

I would rewrite it to just say:

"including most dinosaurs".

Incidentally, this endocast trend would extend down to the base of Maniraptora, as even tyrannosaur brains don’t fill the endocranial cavity.

Line 101: "does" is misspelled.

Lines 101–102: The section reading:

"Hopson 1979); therefore, it should be considered that endocasts do not provide"

I would rewrite it to be a bit more concise such as:

"Hopson 1979). Thus, endocasts do not provide"

Line 103: Delete "among these animals" and just end the sentence after "morphology".

Lines 104–105: I would rewrite this sentence to make it clear that endocasts are still important even if they are limited. Something like:

"Despite these limitations, endocasts still provide the best first-hand information on brain size and shape in extinct species."

Line 109: Delete "that was".

Line 116: Change "size" to "length".

Line 119: I would rewrite this sentence to make it flow a bit better. Something along the lines of:

"Following this, we use the body mass estimates for BSP1964 I458 (4963.6 kg) from Seebacher (2001), as an equivalent estimate for our two specimens."

Lines 143–144: I would change "mean hearing" to "mean best hearing" as described by Walsh et al. 2009.

Line 151: I’m confused by the statement: "As other dinosaurs". Does this refer to specific specimens? Was this meant to read "As with other dinosaurs"? If the latter, can we have a citation or two to verify the claim?

Line 157: Delete "the" in front of "Pachyrhinosaurus lakustai".

Lines 161–162: The last sentence reads strangely. I would rewrite it to something like:

"We also were unable to observe the optic tectum in our specimens."

Lines 182–186: This last section has a fair bit of redundant phrasing in it. As with lines 55 and 69, we can drop the extra modifiers here. You can compress things down a bit too without losing the meaning. For example:

"appear from different foramina, as is seen in other chasmosaurines such as Anchiceratops (Hopson 1979). This is in contrast to centrosaurines like Pachyrhinosaurus, which show the two trigeminal nerve trunks branching from the endocast in closer association (Witmer & Ridgely, 2008; Tykoski & Fiorillo, 2012)."

Line 207: To maintain consistency with the anatomical terminology, I would remove "just below the" and replace it with: "ventral to the".

Lines 219–220: Can rewrite the section after "Therefore" to say: "we assume both Triceratops specimens were around the same ontogenetic stage."

Line 229: Zelentisky is misspelled.

Line 231: Change "theropods and archosaurs" to "theropods and other archosaurs".

Lines 231–234: I would just state your estimated mass from your specimens. You already covered how you came to this mass estimate in your methods section. No need to repeat it here.

Line 264: I would remove "non-avian". The rest of the text uses dinosaurs in a paraphyletic context. Best to stay consistent.

Line 275: Change "Therefore" to "Similarly".

Line 280: Delete "Hence the" and just start the sentence from "CL".

Line 281: Replace "Hence" with "As".

Lines 279–284: I think this section would benefit from a figure call to justify the statement about hearing, or a mention of the range of best hearing for the specimens / species.

Line 283–284: Replace: "have had the ability to hear very distant sounds" with: "have been sensitive to more distant sounds".

Line 296: I would rewrite the start of the paragraph to make it more concise. Suggest:

"Based on our interpretations of the endocranial anatomy of Triceratops, we suggest that (1)"

Line 299: Would remove the bit about "pecking at grass". That, or replace with "pecking at flowers" (grasses not being prevalent at the time).

Line 300: You could expand some on #4 and state that evidence for low gaze stabilization suggests that rapid head movements were uncommon.

Figure 1 caption: Abducens nerve canal is misspelled.

Figure 7: Delete "non-avian".

Experimental design

There are some issues / questions about the methodology.

The olfactory ratios and mean best hearing abilities that are shown in the tables would greatly benefit from an extra column showing where the authors obtained their data. Some of these references (all?) are mentioned in the Materials and Methods section, but I'm still not certain exactly where some of the raw comparative data came from. I go into more detail on this in the Validity of the findings section.

Validity of the findings

There are some large issues with the reported data and some of the interpretations. These will need to be addressed before publication. In particular, the data presented in Table 3.

Table 3: There is something wrong with some of the values in Table 3. You can’t have negative values for hertz barring certain mathematical abstractions. The reported results are correct for Triceratops, Tyrannosaurus, and Gorgosaurus based on the Walsh et al. 2009 equation, but they are also impossible. I suspect that there was an error in the BCL measurements. However, it’s not possible to say for certain without knowing how the measurements were made and where the original data came from. The table needs either an extra column or footnotes by each taxon name listing where the measurements for each specimen came from. I believe that some of the references are mentioned in the Materials and Methods section (e.g., Evans et al. 2009; Witmer and Ridgely 2009), but at least some of these papers do not provide quantified measurements, so I’m not sure where the authors obtained their data. If they were acquired from the images in those papers, it should be mentioned in the methods.

A separate figure showing the criteria used for CL and BCL would be helpful here. Walsh et al. 2009 talk about their CL criteria, but they do not cover the BCL criteria. I’m assuming Walsh et al. followed in the vein of Radinsky (1985) and used the reptilian equivalent of the basioccipital and basisphenoid (i.e., straight-line distance from basal tuber to basipterygoid process as was done by Chure and Madsen 1996). If we go by those limitations, I get a BCL length closer to 100–150 mm for FPDM-V-9775, based on the supplied images. This results in a hearing range around 1530–948 Hz. However, if I use the equation from Gleich et al. 2005 (the one who started this whole trend), I obtain a best hearing frequency of 65 Hz.

Lines 110–111: The authors state that Anderson 1999 found a significant correlation between occipital condyle area and total skull length, but that’s not true. As Anderson 1999 wrote:

"Figure 5B shows the result, and indeed the r2 value (r2=0.732) is increased to a level comparable to that of B. vittatus. However, this result must be interpreted cautiously as the sample size is small (n =5) and the regression is not statistically significant (p ~0.064)."

If this is the best method available right now for ceratopsian head size then so be it, but the authors should mention the limitations of the Anderson equation and that their skull sizes are only rough estimates.


Lines 125–127: The authors used the olfactory ratio calculation method of Zelenitstky et al. 2009. This is another study that has important caveats associated with it that I think the authors need to mention in their study. Namely, Zelenitsky et al. focused only on theropods. Among extant archosaurs, they used a wealth of bird studies and only 3 alligator specimens. The project was very theropod focused and it showed in their results, as their alligator data all fell well above their regression line confidence intervals (though, to be fair, so did their tyrannosaur and dromaeosaur data). It’s likely that the equation that Zelenitsky et al. used is really only good for theropods. The results of this current study further corroborate this as nearly all the non-theropods fell well away from the regression line. I’m fine with the use of the Zelenitsky et al. study (especially since there is not much else out there), but I would make it clear to the audience that the equation used may not be relevant outside of Theropoda.

Lines 235–244: As with lines 125–127, I would stay cautious of interpretations of the results here, as the Zelenitsky et al. 2009 paper was very theropod focused.

Lines 285–287: I would be very careful about using the correlations from Walsh et al. 2009 like this. The authors did find a significant relationship between cochlear length and group size, but their r2 was very low (0.485), indicating that cochlear length is a poor predictor of group size. Low r2s are seen in all the Walsh et al. equations. As such, I would be cautious about any interpretations based on their formulae.


Extra references used in this review

Chure DJ, Madsen JH. 1996. Variation in aspects of the tympanic pneumatic system in a population of Allosaurus fragilis from the Morrison Formation (Upper Jurassic). JVP. 16(1):63-66.

Gleich O, Dooling RJ, Manley GA. 2005. Audiogram, body mass, and basilar papilla length: Correlations in birds and predictions for extinct archosaurs. Naturwissenschaften 92:595–598.

Radinsky, LB. 1985. Approaches in evolutionary morphology: A search for patterns. Ann Rev Ecol Syst. 16:1–14.

Additional comments

I think this is a solid paper that will contribute nicely to our body of knowledge on Triceratops and ceratopsians in general. I appreciate the push for quantification of the data as opposed to the more qualitative approaches done in the past. However, I do think that there are some problems with some of the reported data on hearing frequency that will need to be addressed before the manuscript can continue. Nothing requested should be hard to fix, and I don't suspect a long delay to publication.

Reviewer 3 ·

Basic reporting

The draft is complete in structure and relatively easy to understand, but there are still some grammatical issues and ambiguous parts which need to be corrected or supplemented, especially the sections like ABSTRACT, RESULTS and DISCUSSION, please see the attached PDF.

Experimental design

This is a pioneering study on the quantitative analyses of the endocasts of ceratopsian dinosaurs, although the figures and tables are exquisite, the draft is not detailed enough, please add some information listed in the annotated file.

Validity of the findings

A few views or speculations don't make sense, some analogy are not very accurate, please explain them respectively.

Additional comments

Sorry I didn't revise the draft very carefully because the global virus epidemic, I hope all the people and countries can get through this incident safely. I have annotated some of my opinions in the PDF, and I would recommend publication of this manuscript after major revision.

Annotated reviews are not available for download in order to protect the identity of reviewers who chose to remain anonymous.

---

## Round 0.2 · Minor Revisions

The reviewers have suggested a few minor revisions. One in particular, that you add caveats around the use of ratios, should definitely be made. Once you're able to complete them, we can move the paper forward.

Thanks for sending in such an interesting paper!

·

Basic reporting

No comment.

Experimental design

No comment.

Validity of the findings

No comment.

Additional comments

Thank you to the authors for their thorough revision and accompanying explanation on the latest version of this manuscript. They have done a good job of addressing the major issues, and the paper is much improved. I have only a few minor follow-up comments/suggestions. I do not need to see the revisions, if any, for another round of review.

For FPDM-V-9677, can you indicate how well the estimated skull length matches with the reconstructed skull length? I recognize that the skull is missing some parts, but it would be a helpful check to see if it's consistent with the estimate from occipital condyle size.

Line 254: I would be cautious on assuming they are the same ontogenetic stage. This is potentially true, but alternatively they could be different stages with just the same braincase size. Given size, they're at least probably late subadult (using the terms from Horner and Goodwin 2006), but that's about as far as I would go with the evidence at hand. That said, I don't think this point has a major impact on the results or interpretations.

Line 280-281: Note that Marugán-Lobón et al. 2013 specifically expressed caution on interpreting head posture from semicircular canal position (see third paragraph of the discussion there, and other papers cited therein).

Regarding issues of allometry (e.g., accuracy of predictions for hearing frequency), I would suggest at least acknowledging the breadth of the potential confidence interval for these estimates once you are outside of the size range of the original data used to calculate the equation. Additionally, I would also suggest a brief acknowledgment of the complicating effects of body size in the analysis presented in Figure 6, and the use of ratios. I am still uneasy about how body size and locomotion/posture may be conflated here, although as long as the authors mention this possibility in the text as a possible future research avenue, I am OK with leaving it at that.

·

Basic reporting

No comment

Experimental design

No comment

Validity of the findings

No comment

Additional comments

The authors did a commendable job improving the manuscript and I think it is ready for publication. The only thing that I would recommend fixing is in line 54, which says P. Amitabha. The species name just needs to be put in lowercase (P. amitabha).

Reviewer 3 ·

Basic reporting

no comment

Experimental design

no comment

Validity of the findings

no comment

Additional comments

I am happy with this work for the second time, which is much improved on the first version, including adding some basic decription and comparision. I feel only a little issues that require ironing out before it meets the baseline for publication, provided these are followed/engaged with, I see no reason why this work should not be published (and the figures are very nice).
I have no other technical problems with the manuscript, with my remaining concerns regarding the written English at some points, as well as some content from the response to reviewers that would merit a place in the manuscript. These requirements are not exhaustive and should not take long. I also wish to thanks the authors for the comprehensive responses to the previous reviewers' comments.

---

## Round 0.3 · Minor Revisions

Thank you for your submission and your careful attention to reviewer comments. As a result, I did not send it back to review. In a final read through, I did catch a few grammar issues that should be fixed before publishing. Please make sure they are corrected and then I will accept the paper and move it forward to the proof stage. Thanks for your interesting paper!

Line 60: delete ‘technique’

Line 86: Delete ‘technique’

Line 104: ‘are larger’

Line 109: two spaces between ‘Anchiceratops’ and ‘with’

Line 178: ‘considered that this elongation’

Line 212: two spaces between ‘9775’ and ‘is’

Line 292: two spaces between ‘regression’ and ‘equation’

Line 295: Change to: ‘Therefore the body mass estimate for BSP1964 I458 (4963.6 kg) by Seebacher (2001) was used as the best body mass estimate for our specimens.’

Line 466: ‘via a literature search’

Line 473: ‘literature’, not literatures

Line 750: Please rewrite ‘we will focus mainly on cranial nerves only here’. Mainly and only seem contradictory and I don’t want to change this because I’m afraid I will affect the meaning.

Line 800: Extra space at the end of the sentence ‘(Hopson, 1979) .’

Line 896: ‘variable intra- and interspecifically,’

Line 900: Capitalize Procrustes

Line 923: ‘literature’, not literatures

Line 1171: ‘sensitive to sounds from long distances’

---

## Round 0.4 · accepted · Accept

Thank you so much for your quick resubmission and careful attention to the comments. I am happy to send this paper on!

One thing to note in the proofs: On line 186, there are two periods. Just in case this is not caught by the copy editors, double-check it to make sure it doesn't get published.

Thanks again! I very much enjoyed reading your work.